# WeightCLIP: Aligning Datasets and Models for Weight Space Learning

Aron Asefaw [1]   Konstantinos Tzevelekakis [1]   Damian Falk [1]   Léo Meynent [1]   Damian Borth [1]

## Abstract

Weight space learning aims to learn representations of neural network (NN) weights, enabling different downstream tasks. Existing approaches show promising performance, but lacking a way to shape these weight-space representations using information about the datasets the models were trained on, thus limiting downstream applications. We propose **WeightCLIP**, a method for learning a dataset-aligned latent space for neural networks, where datasets information is induced during training. The NNs are encoded as latent representations using an autoencoder, while dataset samples are encoded using a dataset encoder. The two representations are aligned using a contrastive objective, effectively reshaping the weight-space representations according to the datasets. We demonstrate that such representations can be used for different downstream tasks, including mapping dataset information to a weight-space representation that decode to strong models. In addition, we introduce a latent refinement process for generating models that outperforms standard fine-tuning. Overall, our results demonstrate that explicitly incorporating dataset information improves what can be achieved with weight-space representations across retrieval, generation, and refinement. Code will be available at github.com/HSG-AIML/WeightCLIP.

## 1. Introduction

In recent years, a new training paradigm coined weight space learning (WSL) has emerged that treats the parameters of neural networks as a data modality for representation learning. Once such representations are learned from a population of neural networks (referred to as model zoos),

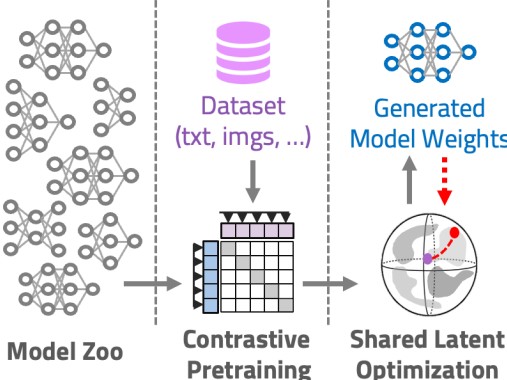

*Figure 1.* WeightCLIP trains a latent representation of model weights using a contrastive loss against the dataset the model weights have been trained on. Once trained, a *data prompt* for an unseen dataset can be used to generate model weights specifically for this dataset.

they can be used for several downstream tasks such as the prediction of model properties from weights (Unterthiner et al., 2020; Eilertsen et al., 2020; Martin & Mahoney, 2021; Schürholt et al., 2021; 2024; Navon et al., 2023; Zhou et al., 2023a), or the synthesis of unseen weights to generate entire models (Schürholt et al., 2022a; 2024; Knyazev et al., 2023; 2025; Kofinas et al., 2024; Wang et al., 2024; 2025; Bedionita et al., 2025b;a; Falk et al., 2025; Liang et al., 2026).

Much like NLP and CV benefited from training on large corpora, WSL has gained substantially from scaling to larger and more diverse model populations (Knyazev et al., 2023; Schürholt et al., 2024; Wang et al., 2024; 2025; Bedionita et al., 2025b;a; Falk et al., 2025). However, while modern NLP and CV increasingly rely on cross-modal supervision (e.g., text–image alignment) to endow representations with shared semantics, weight-space representations often lack an explicit *semantic reference frame* that makes learned representations intuitive to prompt, navigate, and explore. A key reason is that the geometry induced by training is not well-behaved. Model weights do contain information about the data distribution that shaped them, but this signal competes with confounders such as optimization noise, hyperparameter variation, and weight symmetries. As a result, latent spaces learned purely for reconstruction (Schürholt et al., 2022a; 2024; Bedionita et al., 2025b;a) or property prediction (Unterthiner et al., 2020; Eilertsen et al., 2020; Martin & Mahoney, 2021; Schürholt et al., 2021; 2024; Navon et al.,

---

[1]School of Computer Science, University of St.Gallen, Switzerland. Correspondence to: Aron Asefaw <aron.asefaw@unisg.ch>.

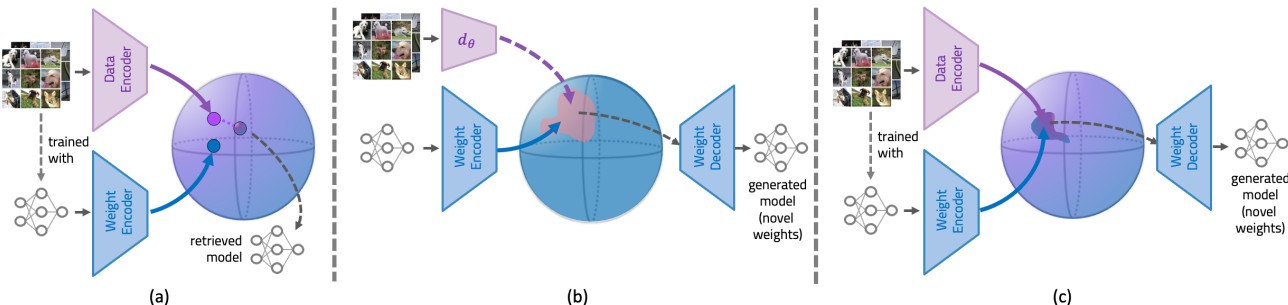

*Figure 2.* The proposed methods differ from other data prompting techniques in WSL. **(a)** While TANS (Jeong et al., 2021) trains with a contrastive objective on encoded model weights and dataset characteristics, the method aims to retrieve known nearest neighbour models. **(b)** In contrast, D2NWG (Bedionita et al., 2025b) trains a generative model able to sample novel weights following a conditioned projection from a dataset space to the frozen weight space representation. **(c)** **WeightCLIP** proposes to train a weight space representation with a contrastive objective on both model weights and dataset characteristics to learn an aligned weight space representation using the dataset as a semantic reference frame, improving prompting, navigation, and refinement capabilities of the learned representation.

2023; Zhou et al., 2023a) can be useful while remaining difficult to navigate: proximity does not have an explicit semantic interpretation, and "navigation in latent space" does not naturally correspond to dataset-level variation. This limits some capabilities we would like WSL to encapsulate such as zero-shot or few-shot transfer, latent space exploration, and controllable weight generation amongst many others.

In this work, we introduce **WeightCLIP** and argue that for model populations, the datasets themselves provide a natural supervision signal analogous to how language provides supervision for vision-language learning. Similar to images on the web, trained models come with an implicit "caption": the data they were trained on. If one could align dataset representations with weight space representations in a shared latent space, then a dataset could provide orientation and act as a *reference* for different downstream tasks. Concretely, given only a small subset of examples from a target dataset, we would like to (i) retrieve models whose training data distribution is the closest, (ii) map a dataset prompt into a weight-space representation that can be decoded into a full model, and (iii) refine this representation using limited target data during test-time, all without expensive sampling in weight space (Wang et al., 2024; 2025; Bedionita et al., 2025b;a).

This perspective turns dataset-model interaction into a cross-modal alignment problem. Contrastive learning has proven particularly effective at aligning heterogeneous modalities into a common space in which similarity is directly measurable and transferable capabilities emerge. However, existing *dataset-model alignment* methods typically use alignment as a learned similarity proxy (for retrieving known models) (Chen et al., 2021; Jeong et al., 2021) or as conditioning for generation (Nava et al., 2023; Tian et al., 2025; Akinwande et al., 2024; Bedionita et al., 2025a;b), while leaving the underlying latent space largely unchanged. Consequently, these approaches do not yield a *dataset-aligned* model manifold: the learned space may support nearest-

neighbor search, but does not become a reference frame for controllable prompting, interpolation, and optimization. On the other hand, a latent space specifically trained to align weights and dataset representations will naturally unlock these capabilities. Additionally, because the contrastive training setup concentrates representations on an approximately hyperspherical shell (Wang & Isola, 2020; Schürholt et al., 2022b), we can efficiently perform *latent-space refinement*: optimize task loss through the decoder while constraining updates to remain near the learned model manifold. This yields a form of latent refinement during test-time that is structurally different to fine-tuning of model weights.

Empirically, we show that dataset-model alignment organizes the learned latent according to the model's training data, improves dataset-to-model retrieval, enables dataset-to-model prompting that generalizes to held-out datasets, and supports efficient latent refinement that can outperform common fine-tuning of generated model weights. Summarizing we make the following contributions:

- **Dataset-aligned Latent Space.** We propose to align the weight space representations by using datasets as reference points, to support: model retrieval, generation of models from data prompts and refinement in the latent space.

- **Dataset-guided NN weights generation.** We propose dataset-to-model mapping methods which can transform a data embedding (i.e., *data prompt*) into a sequence of model embedding, which can be decoded into NN weights for that specific dataset.

- **Refinement on the latent manifold.** We propose a refinement procedure that optimizes task loss through the decoder while constraining latents to a hyperspherical shell and a local neighborhood, enabling effective test-time adaptation in the aligned latent space.

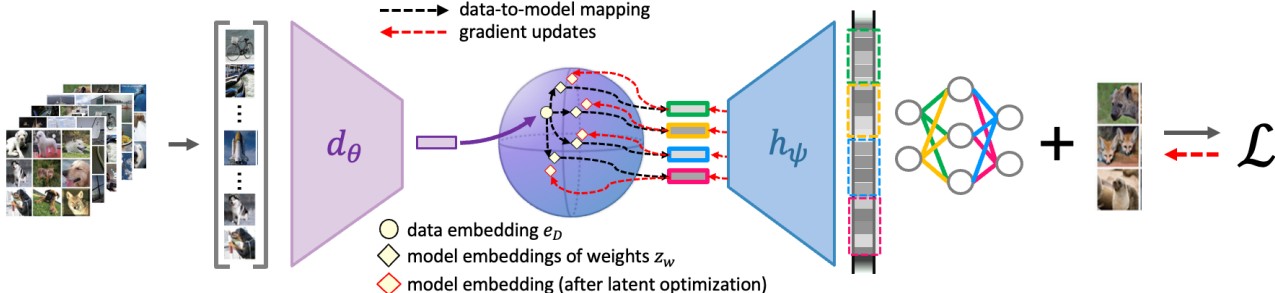

*Figure 3.* After training, the process to generate and refine models weights is done as following: a data encoder $d_\theta$ encodes a subset of the dataset into an data embedding $e_D$ (the *data prompt*) into the aligned latent space. This data embedding is mapped to a sequence of model embeddings $Z$ from which model weights $W$ are decoded using the decoder $h_\psi$ of the trained autoencoder. This newly generated NN is used to evaluated a small set of data samples during test-time. The resulting loss $\mathcal{L}$ is backpropagated through the decoder $h_\psi$ to refine the position of model embeddings $Z$ leading to an improved version of decoded model weights $W$.

## 2. Related Work

We focus this section on alignment-based approaches most directly connected to our method. A broader discussion of weight-space learning is provided in App A.1.

**Multi modal alignment** Contrastive learning (Oord et al., 2018; Wu et al., 2018; Chen et al., 2020; He et al., 2020; Khosla et al., 2020) has become a standard tool for learning semantically structured representations, and has been very effective for aligning heterogeneous modalities into a shared embedding space, most prominently in vision–language representation learning (Radford et al., 2021; Jia et al., 2021; Li et al., 2021; 2022; Masry et al., 2025). By making cross-modal similarity directly measurable in these aligned spaces, semantic retrieval and zero-shot transfer are enabled (Radford et al., 2021; Jia et al., 2021; Li et al., 2021). Beyond retrieval, aligned vision–language latent spaces have been used as controllable interfaces for generation and editing, e.g., by steering synthesis with CLIP-based guidance or CLIP latent spaces and by combining such signals with diffusion or latent-diffusion generators (Patashnik et al., 2021; Ramesh et al., 2022; Nichol et al., 2022; Ho et al., 2020; Rombach et al., 2022). From a geometric standpoint, contrastive alignment induces structured latent geometries, concentrating representations on approximately hyperspherical shells (Wang & Isola, 2020). Another line of work is manifold alignment which studies how representations from different domains can be aligned in a shared space while preserving intrinsic structure (Ham et al., 2004; Wang & Mahadevan, 2009a;b; Xiong et al., 2007). We transfer these alignment principles to weight space learning by aligning dataset and model representations, exploiting the induced geometry for model mapping and refinement.

**Alignment for retrieval and weight generation** Recent work has shown that dataset or task metadata can be inferred from model weights (Putterman et al., 2025; Horwitz et al., 2025). ProbeX (Horwitz et al., 2025), for example, aligns weight embeddings with text embeddings for zero-shot model classification, one-class classification and model retrieval. These works support our premise, but mainly study the direction from weights to dataset or task. Whereas we use dataset examples as prompts for the inverse direction and are able to retrieve, generate, and refine model weights.

A line of work more aligned with our objective explicitly aligns datasets/tasks with models or architectures to enable retrieval, transfer, or weight generation. In Neural Architecture Search (NAS) contrastive objectives have been used to learn architecture comparators and task-adaptive selection in a learned space (Chen et al., 2021; Jeong et al., 2021). A representative example is TANS which learns an aligned embedding space for datasets and models which is used for similarity scoring for task-adaptive retrieval from model zoos (Jeong et al., 2021). Beyond retrieval, recent approaches generate task-adapted parameters using diffusion and guidance mechanisms (Nava et al., 2023; Bedionita et al., 2025b; Jin et al., 2024; Bedionita et al., 2025a). Multiple works (Nava et al., 2023; Tian et al., 2025; Akinwande et al., 2024) uses text-conditioned CLIP guidance to generate small adapter modules for frozen networks, aligning weight encoding to a fixed language embedding space. However, these methods use alignment as external guidance, while keeping the underlying model representation largely unchanged (Bedionita et al., 2025b;a). Hypernetworks are also relevant in this context, since they map conditioning signals directly to model parameters (Ha et al., 2017; Amosy et al., 2024).

While these methods relate datasets and model weights in different ways, they typically use alignment as similarity scoring, external guidance or direct conditional weight generation. On the contrary our approach uses it to reshape the weight-space representations around datasets, allowing datasets to become a reference frame for multiple different downstream tasks.

## 3. Method

**WeightCLIP** aims to make datasets an explicit *reference points* of the latent space of NN weights by aligning dataset and weight-space representations. To this end, the main components of our approach are: an autoencoder for NN weights, a dataset encoder, and a dataset-to-model latent mapper. We initially align the representations of both encoders through a contrastive alignment objective that directly reshapes the autoencoder's latent space, rather than learning a separate projection space as is common in other works (Fig. 2). Intuitively, this pushes the latent space to organize around dataset information such that models trained on similar datasets are represented more similarly. We exploit this structure through a dataset-to-model mapper and refinement procedures to obtain weight-space representations we can decode into well performing model weights (Fig. 3).

### 3.1. Problem formulation

Given small subsets of datasets $D$ and collections of trained NNs, a dataset encoder $d_\theta$ maps datasets to embeddings $e_D \in \mathbb{R}^d$ following the work of (Zaheer et al., 2017), and a weight-space autoencoder compresses NN weights $\boldsymbol{W}$ to latent representations using an encoder $g_\phi$, and reconstructs them with a decoder $h_\psi$. Where $d$ is the latent dimension for both weight and dataset representations, $\{\phi, \psi\}$ the parameters of the weight-space autoencoder and $\theta$ the parameters of the dataset encoder. In this setting, we want to learn a shared representation space in which a dataset can provide orientation and act as a reference point for navigating this aligned space, ultimately leading to well-performing model representations for that dataset.

This setup is intended for an amortized transfer setting. The model zoo, weight autoencoder, dataset encoder, and mapper are constructed once, after which new target datasets only require prompt encoding, mapping, decoding, and optionally short adaptation. Thus, the upfront cost of learning the aligned latent space is amortized over many downstream datasets for a known architecture family.

### 3.2. Processing neural network weights

Following the work of Schürholt et al. (2024), we represent NNs as sequences of weight tokens, where each token, of size $t_k$, is obtained by flattening a contiguous subset of layer parameters. More specifically, we reshape weights into 2D matrices, slice them row-wise along the outgoing channel, pad to $t_k$, and concatenate them across layers forming a token sequence we call window. The autoencoder processes model parameters per window. As a result, the encoder produces a window of $L$ token representations $\boldsymbol{z}_W \in \mathbb{R}^d$, where $L$ corresponds to the window size. Notably, for small networks, a single window may span the full parameter set, whilst larger models are partitioned into multiple windows.

### 3.3. Reconstruction and alignment supervision

The weight space autoencoder $\{g_\phi, h_\psi\}$ and the dataset encoder $d_\theta$ are optimized in parallel using different supervision objectives. For the weight space autoencoder a composite loss is used combining a model weights reconstruction loss and a contrastive dataset alignment loss. The complete training objective of the autoencoder is:

$$\mathcal{L}_{\text{auto}} = \mathcal{L}_{\text{recon}} + \mathcal{L}_{\text{align}} \tag{1}$$

On the other hand, the dataset encoder is supervised using the dataset alignment loss $\mathcal{L}_{\text{align}}$, as well as a classification loss $\mathcal{L}_{\text{ce}}$ by attaching a classification head which predicts the dataset identity from the dataset embedding $e_D$:

$$\mathcal{L}_{\text{data}} = \mathcal{L}_{\text{align}} + \mathcal{L}_{\text{ce}}. \tag{2}$$

**Reconstruction loss** Following the related literature, we use the difference between the original and reconstructed weights as a supervision signal. With $B$ denoting the batch ingested, the reconstruction loss is formulated as:

$$\mathcal{L}_{\text{recon}} = \frac{1}{B} \sum_{i=1}^{B} \|h_\psi(g_\phi(\boldsymbol{W}_i)) - \boldsymbol{W}_i\|_2^2 \tag{3}$$

**Dataset alignment loss** To align the model and dataset representations we adopt a token-level contrastive alignment loss. Following the hypothesis that dataset signal is distributed uniformly across all the token representations of the model, which we empirically verify in App B.2, we don't handle tokens differently based on their position in the model. Thus, given a batch of windows, each of them with token representations $\boldsymbol{Z}^{(i)} = [\boldsymbol{z}_1^{(i)}, \ldots, \boldsymbol{z}_L^{(i)}]$ and a dataset embedding $e_D^{(i)}$, we align them using a contrastive objective:

$$\mathcal{L}_{\text{m}\to\text{d}} = -\frac{1}{BL} \sum_{i=1}^{B} \sum_{t=1}^{L} \log \frac{\exp(\tilde{s}(\boldsymbol{z}_t^{(i)}, e_D^{(i)}))}{\sum_{j=1}^{B} \exp(\tilde{s}(\boldsymbol{z}_t^{(i)}, e_D^{(j)}))} \tag{4}$$

where $\tilde{s}$ stands for the cosine similarity with temperature $\tilde{s}(\boldsymbol{a}, \boldsymbol{b}) = \tau^{-1} \left(\boldsymbol{a}^\top \boldsymbol{b} \,/\, \|\boldsymbol{a}\|\|\boldsymbol{b}\|\right)$, and $\text{m} \to \text{d}$ indicates the direction of the alignment from model to dataset. For stability we also include the reverse direction to define our composite alignment loss: $\mathcal{L}_{\text{align}} = \mathcal{L}_{\text{m}\to\text{d}} + \mathcal{L}_{\text{d}\to\text{m}}$.

### 3.4. Dataset-to-model mapping

Although datasets and models latent spaces are now aligned, the mapping from a dataset prompt to a model representation is not trivial. Firstly, there is a cardinality mismatch between a dataset embedding $e_D \in \mathbb{R}^d$ and a window of token representations $\boldsymbol{Z} \in \mathbb{R}^{L \times d}$. More importantly, there are residual cross-modal discrepancies that prevent the underlying spaces from being completely aligned, as it has been shown by (Mistretta et al., 2025; Liang et al., 2022; D'Orazio et al., 2026). To remedy these problems, we propose learning a lightweight mapper that is able to yield a window of token representations given a dataset embedding.

Notably, by employing such a mapper the model encoder is no longer necessary for model generation as only the dataset encoder, mapper and model decoder are involved.

**Linear mapper** As a minimal baseline, we regress the *entire* token sequence from the dataset prompt using a single linear map on the flattened representation. Concretely, we predict a vector in $\mathbb{R}^{L \cdot d}$ via

$$\text{vec}(\hat{\boldsymbol{Z}}) = \boldsymbol{A}\boldsymbol{e_D} + \boldsymbol{b} \qquad (5)$$

where $\boldsymbol{A} \in \mathbb{R}^{(L \cdot d) \times d}$ and $\boldsymbol{b} \in \mathbb{R}^{L \cdot d}$ are learnable parameters and $\text{vec}(\cdot)$ denotes flattening. We then reshape $\text{vec}(\hat{\boldsymbol{Z}})$ into $\hat{\boldsymbol{Z}} \in \mathbb{R}^{L \times d}$. Despite operating on the full sequence, this baseline remains computationally cheap (a single affine layer) and tests how far alignment alone makes the model latent representation predictable from dataset semantics. We train $(\boldsymbol{A}, \boldsymbol{b})$ on pairs $\{\boldsymbol{e_D}, \boldsymbol{Z}\}$ from our training model zoos, using ridge regression with a mean-squared-error loss.

**Memory-bank mapper** It has been shown in prior work that enforcing some bias to known priors can give a performance increase in cross-modal alignment (Masry et al., 2025). Following this we bias predictions toward the manifold of real model representations by introducing a memory bank $\mathcal{M} = \{\boldsymbol{Z}^{(m)}\}_{m=1}^{N}$ of token representations from the training models. Given a dataset prompt $\boldsymbol{e_D}$, we replicate it across token positions and add a positional encoding $\text{PE}(t)$ to form token-conditioned inputs. A predictor $s_{\varphi}$ then outputs logits for each token position $t$ over memory-bank indices $m$:

$$\boldsymbol{\alpha}_t = \text{softmax}(s_{\varphi}(\boldsymbol{e_D} + \text{PE}(t))) \in \mathbb{R}^N \qquad (6)$$

We denote the $m$-th component by $\alpha_t^{(m)}$ and synthesize tokens via convex combination:

$$\hat{\boldsymbol{z}_t} = \sum_{m=1}^{N} \alpha_t^{(m)} \boldsymbol{z_t^{(m)}} \qquad (7)$$

The mapper is trained using a soft neighborhood-based supervision that reflects in model-latent space (see App C.6).

### 3.5. Refinement in latent space

After mapping, models often exhibit classifier head mismatches due to differing numbers of output classes, making a refinement process necessary at test time. A standard baseline is fine-tuning in weight space. We additionally propose a refinement directly in latent space by exploiting the learned latent geometry to remain on (or near) the correct model manifold. This is achieved through a two stage process done at test time.

**Stage 1: Global prior** Prior work (Schürholt et al., 2022a) has empirically shown that the model embeddings concentrate around a hyper-spherical shell, we apply this constraint per-token. Let $\boldsymbol{c} \in \mathbb{R}^d$ and $r$ denote the average token mean

and norm, respectively, estimated from training data. We define the per-token projection operator as:

$$\Pi_{\text{shell}}(\boldsymbol{z_t}) = \boldsymbol{c} + r\frac{\boldsymbol{z_t} - \boldsymbol{c}}{\|\boldsymbol{z_t} - \boldsymbol{c}\|_2}, \quad t = 1, \dots, L \qquad (8)$$

Which rescales a latent to lie on this shell. We apply $\Pi_{\text{shell}}$ to a sequence of tokens $\boldsymbol{Z}$ by applying the projection to each token. Given an initial latent sequence $\boldsymbol{Z_0}$ from a mapper we can now initialize a refinement process from:

$$\boldsymbol{Z}^{(0)} = \Pi_{\text{shell}}(\boldsymbol{Z_0}) \qquad (9)$$

ensuring that optimization begins in a region of latent space corresponding to valid trained models.

**Stage 2: Local optimization in latent space** We then perform a local optimization which treats the full window $\boldsymbol{Z} \in \mathbb{R}^{L \times d}$ as the optimization variable and decreases loss on target data by backpropagating through the decoder. Specifically, we decode weights from $\boldsymbol{Z}$ using the decoder $h_{\psi}$ and evaluate the resulting model on a small set of target data from $D$'s training set. In the case of models spanning multiple windows, this optimization is applied jointly to all windows. The refinement objective is:

$$\mathcal{L}_{\text{ref}}(\boldsymbol{Z}) = \mathcal{L}_{\text{task}}(h_{\psi}(\boldsymbol{Z})) + \gamma\|\boldsymbol{Z} - \boldsymbol{Z}^{(0)}\|_F^2 \qquad (10)$$

Optimization uses standard gradient descent:

$$\boldsymbol{Z}^{(i+1)} = \boldsymbol{Z}^{(i)} - \eta\nabla_{\boldsymbol{Z}}\mathcal{L}_{\text{ref}}(\boldsymbol{Z}^{(i)}) \qquad (11)$$

**Finetuning and BN conditioning** We perform refinement with an amount of target-data and compare that against standard fine-tuning by using the same number of gradient steps and the same amount of target-data. For the batch-normalization parameters, we apply BN conditioning following prior work (Schürholt et al., 2024; Gupta et al., 2026).

## 4. Experiments

### 4.1. Implementation details

For encoding and decoding weights of NNs we employ the SANE autoencoder (Schürholt et al., 2024). Notably, by operating directly on the token-level representations, our method scales linearly with the number of parameters of the processed architectures, and model-level representation aggregation is not necessary. Furthermore, for the dataset encoder, we are using the DeepSets architecture proposed by Zaheer et al. (2017) that maps subsets of images to dataset embeddings we use as prompts. At test time we translate multiple dataset prompts into models and, similar to subsampling in Schürholt et al. (2024), report the results for the top-$m$ models. Finally, we train our autoencoder and dataset encoder in parallel, as described in Section 3.3. Additional implementation details and hyper-parameters are provided in App (Sec. C).

## 4.2. Experimental setup

**Model zoos** We evaluate our approach on two model zoos, each consisting of a population of trained models sharing a single architecture but trained on multiple datasets. The first zoo includes 10'000 convolutional neural network (CNN) checkpoints. These checkpoints were acquired by training 100 models across 20 distinct image classification datasets spanning diverse visual domains (e.g., medical imaging, aerial imagery, symbols, and objects), following a subset of the dataset collection introduced in TANS (Jeong et al., 2021); see App. C.1. For each training run we keep the weights of five models between the $41^{st}$ and the $45^{th}$ epochs. Unless stated otherwise, all experiments in the paper are conducted on the CNN model zoo. The second model zoo, that is used to showcase the ability of our method to scale up to larger architectures, contains 1'000 ResNet-18 checkpoints collected, in a similar fashion, by training 20 models on 10 distinct datasets, keeping five checkpoints per training run as mentioned above.

**Split of downstream image datasets** To evaluate our approach we keep six downstream datasets to be used exclusively for testing. More specifically, for both of the aforementioned model zoos and for the alignment and mapping components of our method we only use the training image datasets so that the unseen test image datasets are used to evaluate the performance of the generated models.

**Baselines** We compare our approach against TANS (Jeong et al., 2021), the most comparable prior work in terms of problem formulation and use of bidirectional dataset-model alignment. Since TANS is restricted to retrieval we further include a generation baseline using a setting similar to D2NWG (Bedionita et al., 2025b). Where we specifically evaluate against their data projection into the latent space. And to compare against a more direct conditional weight-generation approach we include a hypernetwork baseline adapted from Text2Model (Amosy et al., 2024). In this baseline a DeepSets encoder maps a subset of target images to a single dataset embedding, which conditions a Text2Model-style hypernetwork that directly predicts full-model tokens for a ResNet18 model. See App (Sec. C) for more details. Together, these baselines cover retrieval-only, projection-based generation, and direct conditional weight generation alternatives.

## 4.3. Dataset alignment effect on model representations

In this section, we evaluate the impact of dataset alignment, as discussed in Sec. 3.3, and provide evidence to support two main hypotheses: First, that dataset alignment clusters the model representation space around the training downstream datasets. Second, that dataset prompts can be used to navigate through the dataset aligned model representation space.

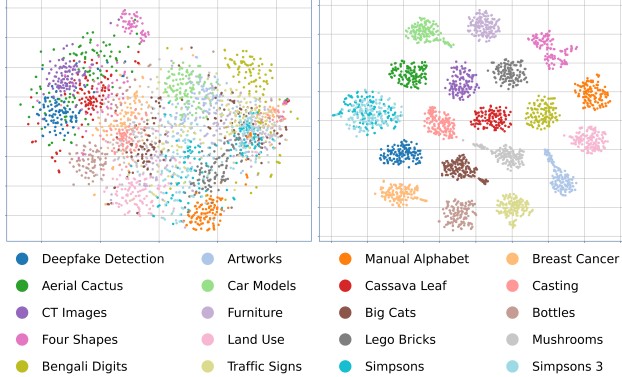

Deepfake Detection · Artworks · Manual Alphabet · Breast Cancer · Aerial Cactus · Car Models · Cassava Leaf · Casting · CT Images · Furniture · Big Cats · Bottles · Four Shapes · Land Use · Lego Bricks · Mushrooms · Bengali Digits · Traffic Signs · Simpsons · Simpsons 3

*Figure 4.* t-SNE projection of NN embeddings from the model zoo, obtained by mean-pooling token representations. Colors indicate the dataset each model was trained on. **Left:** Trained without alignment, where dataset alignment is not explicitly enforced, resulting in substantial overlap between NN embeddings. **Right:** Trained with the proposed alignment, where models trained on the same dataset form more compact and clearly separated clusters.

**Reshaping the model latent with dataset alignment** To validate whether dataset alignment separates the model latent space, we visualize model representations with t-SNE. For each trained NN, we compute a single model representation by averaging its token representations $z_t$. Then, having each point correspond to a model representation, we color each point according to the corresponding training dataset. We compare both aligned and non aligned model encoders and display the result in Fig. 4. As observed in the figure, training without dataset alignment results in a significant overlap between representations of models trained on different datasets. Conversely, dataset alignment leads to tight clusters of model representations, organized by training dataset. A model–dataset similarity matrix in App. D.2 further shows a strong diagonal structure between dataset and model embeddings. To complement our qualitative observations, we use $k$-nearest-neighbor ($k$NN) to classify model representations to their corresponding training datasets. Validating our hypothesis, the use of dataset alignment increases the classification accuracy by 21% (i.e. $77\% \rightarrow 98\%$).

**Navigating model latent using dataset prompts** To evaluate the degree of control dataset prompts have over model generation, we conduct the following experiment. Using dataset prompts from two semantically diverse datasets, we feed their linear interpolation to our mapper component resulting in a model representation that is ultimately decoded to a model, and report its accuracy on both datasets. The accuracy corresponding to these interpolated dataset embeddings, as the interpolation coefficient $t$ varies, are reported in Fig. 5. As $t$ increases, performance on one dataset improves while performance on the other degrades. This indicates that dataset alignment induces dataset related information in the model latent space, enabling navigation through dataset prompts.

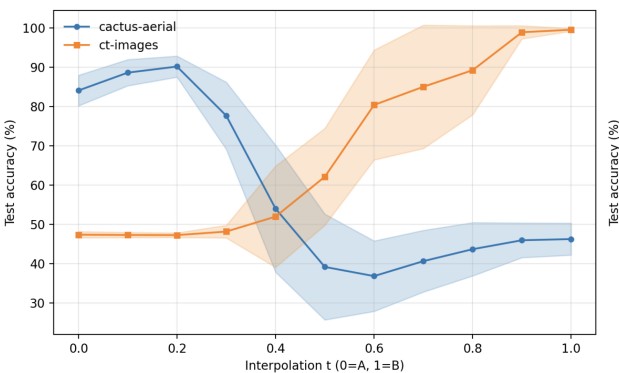

*Figure 5.* Latent-space navigation via dataset prompt interpolation. We linearly interpolate between dataset prompts from two distinct datasets (*cactus aerial → ct images*), map each interpolated prompt to a NN latent representation, and decode it. The plot shows downstream accuracy of the generated models on both datasets as a function of the interpolation coefficient $t$, revealing a smooth trade-off in performance between the two datasets.

## 4.4. Model retrieval using dataset prompts

As a downstream effect of the restructured latent, we explore if alignment enables stronger dataset-to-model retrieval. Given a dataset prompt, we retrieve the nearest model latent from a candidate pool using cosine similarity. Depending on the setting, retrieval is evaluated either by matching the dataset identity of the retrieved model or by measuring its downstream performance.

**In-distribution** We first evaluate in-distribution retrieval performance. Dataset prompts are drawn from the same set of training datasets used during alignment, while the candidate pool consists of held-out model instances from the same model zoos that were not seen during encoder training. Retrieval is considered correct if the retrieved model was trained on the same dataset as the query prompt. Tab. 1 reports recall at different cutoffs for this task. Alignment improves recall over both training without alignment loss and TANS, demonstrating a more queryable latent space.

**Out-of-distribution** We next evaluate retrieval in an out-of-distribution setting, where the query dataset was unseen during training. Given an unseen dataset prompt, we retrieve the nearest model from the candidate pool and directly eval-

*Table 1.* In-distribution dataset–model retrieval, expressed with the recall percentage. A dataset prompt is used to query a pool of model latents, and retrieval is considered correct if a model trained on the same dataset appears among the top-$K$ results (R@K). WeightCLIP improves retrieval over training without alignment loss and also outperforms TANS.

| Method | R@1 | R@5 | R@10 |
|---|---|---|---|
| TANS | 90 | 100 | 100 |
| WeightCLIP w/o align. | 10 | 20 | 25 |
| **WeightCLIP** | **95** | **100** | **100** |

*Table 2.* Out-of-distribution retrieval performance of Resnet18 models. We report average target accuracy (%) across the OOD datasets when retrieving the nearest model using each method, evaluated zero-shot and after one epoch of fine-tuning.

| Method | Zero-shot | Ep. 1 |
|---|---|---|
| TANS | 25.8±2.5 | 43.2±3.9 |
| D2NWG (projection)[1] | 26.2±1.7 | 53.9±3.7 |
| **WeightCLIP** | **26.3±2.0** | **55.6±2.1** |

uate its performance on the target dataset. The results are shown in Tab. 2, where our aligned representation obtains the highest average accuracy after one epoch, indicating that our approach induces the most aligned latent spaces among the compared methods.

## 4.5. Model generation using dataset prompts

While retrieval already provides evidence that dataset information is encoded in the latent space of neural networks through neighborhood structure, we next evaluate whether this information can be used to generate new models. Specifically, we study dataset-to-model mapping in the aligned latent spaces. Unlike retrieval, which selects among existing models, generation requires mapping a new model from a dataset prompt, providing a more direct test of whether dataset information have been induced in the latent space of NNs and can be used to generate new networks.

**Dataset-to-model generation** Tab. 3 reports mapping performance on the ResNet18 model zoo, evaluating both zero-shot performance (Ep. 0) as well as short finetuning to handle structural mismatches in the classifier head. Corresponding results on the CNN zoo are provided in App B.1.

Overall, mapping from the aligned latent space consistently outperforms both baselines and training from scratch across evaluation settings. This indicates that alignment makes dataset prompts informative enough to generate full model initializations that are well matched to the target dataset, even for out-of-distribution datasets. With additional fine-tuning the gaps shrink, but generated initializations remain superior.

To understand which components drive this improvement, we compare the different mappers. The linear mapper is already competitive across epochs, suggesting that a simple mapping from dataset prompts to model tokens is often sufficient once the latent space is aligned. The memory-bank mapper provides a complementary inductive bias by constraining predictions to convex combinations of real training latents. This often improves early performance, especially at Ep. 0, and yields the best results for several datasets, with the linear mapper remaining competitive in later fine-tuning stages.

---

[1]Emulation of D2NWG (Bedionita et al., 2025b)

*Table 3.* Dataset-to-model generation on ResNet18 models. Given an OOD dataset prompt, each method predicts a full token sequence of a model and decodes it. We report test accuracy (%) after 0, 1, and 10 epochs of fine-tuning on the target dataset (mean±std over seeds). Where LM denotes the linear mapper and MBM denotes the memory-bank mapper.

| Ep. | Method | Colo.H | COVID-19 | CIFAR10 | Speed. | Honey P. | Real/Draw. |
|---|---|---|---|---|---|---|---|
| | tr. fr. scratch | ~12.5 | ~33.3 | ~10.0 | ~25.0 | ~50.0 | ~10.0 |
| | TANS | 8.5±6.5 | 44.9±0.9 | 8.6±1.8 | 30.6±2.8 | 51.9±1.6 | 10.3±1.3 |
| 0 | Hypernetwork | 12.4±0.0 | 35.6±0.0 | 9.5±0.0 | 13.9±0.0 | **59.7±0.0** | 12.6±0.0 |
| | **WeightCLIP**-LM | 13.7±0.3 | **47.4±0.4** | 10.8±0.1 | 36.7±3.2 | 56.9±1.2 | 16.2±0.9 |
| | **WeightCLIP**-MBM | **18.3±1.8** | 45.9±0.3 | **11.0±0.2** | **40.0±1.0** | 52.6±1.0 | **18.1±0.2** |
| | tr. fr. scratch | 54.1±1.0 | 60.7±4.1 | 38.9±0.6 | 26.1±6.2 | 56.4±1.9 | 27.0±1.2 |
| | TANS | 29.2±9.3 | 73.5±1.9 | 41.5±0.3 | 32.4±1.6 | 60.7±7.7 | 21.9±2.8 |
| 1 | Hypernetwork | 59.1±0.5 | 82.9±0.5 | 42.7±0.4 | 21.1±1.0 | 82.3±3.1 | 34.2±0.3 |
| | **WeightCLIP**-LM | **68.4±4.6** | 84.7±1.4 | 46.2±3.1 | 41.7±6.8 | **82.8±6.4** | **35.9±2.7** |
| | **WeightCLIP**-MBM | 68.4±3.8 | **90.8±0.6** | 47.7±3.5 | **48.0±2.6** | 72.9±3.7 | 35.4±1.7 |
| | tr. fr. scratch | 74.8±5.7 | 81.7±7.1 | 68.1±0.7 | 48.3±8.4 | 84.7±1.2 | 42.6±1.2 |
| | TANS | 71.9±2.3 | 89.0±0.5 | 68.4±0.1 | 51.9±10.5 | 92.6±6.9 | 44.3±2.2 |
| 10 | Hypernetwork | 75.1±0.8 | 90.0±0.3 | **71.5±0.5** | 58.2±1.2 | 92.8±2.3 | 48.7±0.8 |
| | **WeightCLIP**-LM | **84.6±1.3** | 92.6±0.8 | 70.5±4.3 | 70.6±5.4 | **95.3±2.3** | **55.2±1.2** |
| | **WeightCLIP**-MBM | 82.2±1.4 | **95.0±0.3** | 70.3±5.4 | **74.3±2.3** | 94.6±2.8 | 53.1±1.1 |

*Table 4.* Average Ep. 0 performance (%, mean±std) for linear-mapper generation under different alignment configurations on the first three OOD datasets. All settings use the same autoencoder and mapper; only the alignment setup differs.

| Alignment setting | Avg. Acc. |
|---|---|
| Ours w/o alignment | 14.5±4.0 |
| D2NWG (projection)[1] | 23.9±4.0 |
| **WeightCLIP** | **26.6±2.1** |

*Table 5.* Average Ep. 0 performance (%, mean±std) for nearest-neighbor selection and dataset-to-model generation for ResNet18 on the OOD datasets. All methods use the same aligned latent space; only the selection or mapping strategy differs.

| Method | Avg. Acc. |
|---|---|
| Nearest Neighbor | 26.3±2.0 |
| Linear mapper | 30.9±1.0 |
| Memory bank mapper | **31.5±0.8** |

Finally, these trends hold across substantially different architectures, from single-window CNNs to multi-window ResNets, indicating that the proposed alignment remains effective as architectural complexity increases. We interpret dataset-to-model generation as producing dataset-conditioned initializations rather than fully solved zero-shot models. Ep. 0 performance shows that the generated weights are already oriented toward the target task, while the stronger Ep. 1 and Ep. 10 results demonstrate that this provides a useful starting point for rapid adaptation.

**Generation under different alignment settings** To highlight the importance of our alignment component during generation we conduct the following ablation study where we ablate which encoders participate in the alignment objective. Specifically, we consider three settings: (i) training without alignment loss, (ii) our generation baseline D2NWG, where only the dataset encoder is aligned and (iii) alignment of both encoders.

Tab. 4 shows that aligning the dataset encoder alone yields a slight improvement over training without alignment at all, indicating that dataset representations benefit from being shaped towards model space. However the strongest gains are achieved only when the latent space of the NNs is also aligned. This confirms that mapping quality is not driven solely by better dataset prompts but also depends on reshaping the NNs latent space to better reflect dataset information.

### 4.6. Generation vs. nearest-neighbor selection

Nearest-neighbor retrieval tests whether the aligned latent space is queryable, while mapping tests whether it can synthesize model representations beyond selecting an existing zoo entry. We therefore compare both mappers against nearest-neighbor selection in the same latent space. As shown in Tab. 5, both mappers outperform nearest-neighbor selection, indicating that the generated representations are not simply equivalent to the closest retrieved model.

For the memory-bank mapper, the output remains a convex combination of training model tokens, but it is not hard retrieval: each token position can use a different soft mixture over the bank. The gain over nearest-neighbor selection suggests that this token-wise composition goes beyond choosing the closest existing model.

## 4.7. Refinement in the aligned latent space

Finally we evaluate refinement in the latent-space for repairing mismatches after mapping, such as incorrect classifier heads, for the generated models. Unlike weight-space fine-tuning, this procedure updates the decoded model indirectly by optimizing its latent representation, thereby keeping the update constrained by the learned model manifold.

**Refinement compared to finetuning** We compare refinement of the generated models in the aligned latent space to standard weight-space fine-tuning under matched compute budgets. Tab. 6 shows that refinement achieves comparable or superior performance with the same amount of updates and data.

*Table 6.* Refinement in latent space compared to standard fine-tuning, under the same computing budget. All methods start from the same mapped model (*Init*). We use 20 gradient steps with a batch of 100 target images. We report mean±std over 4 models.

| Dataset | Init | Fine-tune | Latent Refine | Δ |
|---|---|---|---|---|
| Colo.H | 23.9±0.1 | 36.7±0.9 | **37.0±0.6** | +0.3 |
| COVID-19 | 46.3±0.8 | 57.3±1.3 | **65.6±2.0** | +8.3 |
| Speed. Signs | 28.7±3.5 | 48.2±1.3 | **60.2±5.7** | +12.0 |
| CIFAR10 | 10.2±1.1 | 16.7±1.4 | **19.5±1.7** | +2.8 |
| Honey P. | 46.3±0.7 | 54.6±0.7 | **62.0±5.1** | +7.4 |
| Real/Draw. | 10.7±0.6 | 19.2±1.6 | **24.9±2.0** | +5.7 |
| **Average** | 27.7±1.1 | 38.8±1.2 | **44.9±2.9** | **+6.1** |

**Decomposing the gains of latent refinement** Tab. 7 decomposes the sources of improvement in the refinement. First, enforcing the hyper-spherical constraint is necessary to obtain gains over fine-tuning. Second, when controlling for this constrain we can see that the aligned initializations substantially outperform non-aligned ones. Taken together this shows that alignment provides a structured latent space enabling not only generation but also the refinement of weight space representations.

*Table 7.* Ablation of refinement in the latent space. We vary whether refinement is constrained to the hyperspherical shell and whether the latent is aligned. All refinement settings use identical optimization steps and data and are averaged across datasets.

| Method | With shell | Without shell |
|---|---|---|
| Standard fine-tuning | – | 38.8 |
| WeightCLIP w/o align. | 40.7 | 37.2 |
| **WeightCLIP** | **44.9** | 42.4 |

## 4.8. Limitations

In this work, we demonstrate that cross-modal alignment between dataset embeddings and weights latent representations is possible, and that it improves the capabilities of the WSL backbone by making its latent space more structured. We do recognize, however, that our exploration comes with some limitations in scope. Our work is limited to computer vision datasets, in line with the related work: the possibility to align dataset and weights representations across modalities (e.g., vision and language) remains unexplored. Furthermore, our approach limits itself to models of the same architecture: if we were to apply our method to heterogeneous model zoos such as in Falk et al. (2025); Hanna et al. (2026), it is unclear how the latent should be structured to represent both the underlying dataset and architecture; it is also unclear whether our hyper-spherical prior for latent refinement would hold in that case. A further limitation is that our architecture does not explicitly enforce permutation or other weight-space symmetries. Instead, following recent generative weight-space approaches, we rely on training augmentations to learn representations that are useful despite these symmetries (Schürholt et al., 2024).

## 5. Conclusion

Our work shows that dataset information can be used to reshape NNs representation space to enable model retrieval, generation and refinement of models for targeted dataset distributions, addressing a key limitation of existing WSL methods. To that end, we introduce a contrastive alignment objective to align dataset and weight space representations. We further employ a simple dataset-to-model mapper to translate dataset prompts to model representations, ultimately enabling the generation of NN weights. These are further refined in the latent space under hyper-spherical shell constraint that is shown to match or exceed common finetuning. Through a range of experiments, we demonstrate that alignment results in the clustering of weight space representations, significantly improving dataset-to-model retrieval, and generation of strong model initializations on both in- and out-of-distribution datasets, outperforming previous works and training from scratch.

More broadly, our work is an important contribution to the field of weight space learning: by allowing the generation of novel model weights from a dataset embedding, our method opens up the possibility to significantly accelerate model training for a known architecture on a new dataset. While future work will have to extend the validity domain of our approach, our paper represents a step towards making NN training vastly more time- and compute-efficient.

## Impact Statement

This paper presents work whose goal is to advance the field of Machine Learning. There are many potential societal consequences of our work, none which we feel must be specifically highlighted here.

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

# A. Additional Related Work

## A.1. Weight-space learning

Representation learning in the space of neural-network weights has grown into a broad field spanning both model analysis and weight generation. Early work focuses on predicting model properties, such as accuracy, directly from weights using learned or engineered weight features (Unterthiner et al., 2020; Eilertsen et al., 2020; Mahoney & Martin, 2019; Martin & Mahoney, 2021; Han et al., 2026; Ettling et al., 2026). Other work studies the structure of trained weights e.g. invariance-equivariance properties, symmetries, and geometry, which motivates permutation-/equivariance-aware representations (Navon et al., 2023; Kofinas et al., 2024; Lim et al., 2024; Zhou et al., 2023a;b; 2024). A prominent direction uses autoencoders to learns weight latent spaces, introducing the term hyper-representations, which enable both discriminative and generative use-cases (Schürholt et al., 2022a; 2024). In parallel, weight generation has been explored with diffusion and other generative models (Wang et al., 2024; Bedionita et al., 2025b; Wang et al., 2025; Jin et al., 2024). Other efforts investigate how model zoo composition and large-scale model collections can impact weight representations and generalization (Falk et al., 2025; Horwitz et al., 2026). Our work builds on the autoencoder presented in (Schürholt et al., 2024) but focuses on explicitly shaping the weight representation space using the dataset information, thus supporting retrieval, mapping, and refinement.

# B. Additional Results

This Appendix provides extended quantitative results and additional analyses that support the main claims. We first report full dataset-to-model mapping tables for the CNN ar-

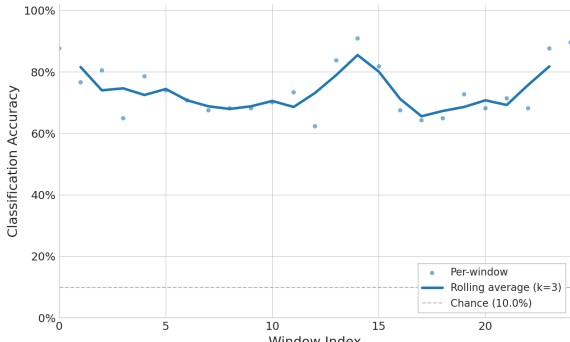

*Figure 6.* Classification accuracy from window aggregated token representations for ResNet models using linear classifiers, with the individual results as well as a rolling average shown. Results are broadly consistent across positions, supporting our token-level supervision.

chitecture, then include auxiliary analyses that probe where the dataset signal lives in the latent and lastly study how sensitive WeightCLIP is to the size used for the dataset prompt.

## B.1. Extended mapping results on CNN models

Tab. 8 reports dataset-to-model mapping on CNN model zoos, evaluated at initialization (Ep. 0) and after 1 and 10 epochs of downstream fine-tuning. It follows the same experimental protocol as for the Resnet18 zoos with the exception of no comparisons to hypernetworks.

## B.2. Dataset signal is distributed across tokens.

To test the assumption underlying token-level alignment mentioned in Section 3.1, we train linear probes to predict dataset identity from window-aggregated embeddings (ResNet). Fig. 6 shows that probe accuracy is broadly comparable across positions, indicating that dataset identity is not strongly localized and that token-level supervision provides consistent learning signal across architectures.

*Table 8.* Dataset-to-model generation on CNN models under the same evaluation protocol as Tab. 3. Given an OOD dataset prompt, each method predicts a full token sequence of a model and decodes it. We report test accuracy (%) after 0, 1, and 10 epochs of fine-tuning on the target dataset (mean±std over seeds). Where LM denotes the linear mapper and MBM denotes the memory-bank mapper.

| Ep. | Method | Colo.H | COVID-19 | Speed. | CIFAR10 | Honey P. | Real/Draw. |
|-----|--------|--------|----------|--------|---------|----------|------------|
| 0 | tr. fr. scratch | ~12.5 | ~33.3 | ~25.0 | ~10.0 | ~50.0 | ~10.0 |
| | TANS | 9.3±4.6 | 35.4±19.9 | 28.7±3.2 | 9.3±0.8 | 49.1±0.8 | 9.5±1.5 |
| | **WeightCLIP**-LM | 20.6±4.3 | 46.3±1.4 | 36.7±1.0 | **12.9±0.7** | **54.4±2.1** | 14.3±1.3 |
| | **WeightCLIP**-MBM | **22.8±2.1** | **46.9±0.2** | **41.5±1.6** | 12.3±0.2 | 49.9±0.6 | **15.4±0.9** |
| 1 | tr. fr. scratch | 54.3±6.0 | 74.3±6.0 | 38.3±6.2 | 46.2±1.4 | 63.6±10.3 | 31.2±1.5 |
| | TANS | 58.9±4.1 | 78.4±5.0 | 45.4±13.1 | 46.2±5.4 | **74.1±10.5** | 34.4±1.8 |
| | **WeightCLIP**-LM | **62.0±1.2** | 82.7±1.7 | 44.6±3.4 | **53.3±1.7** | 70.5±3.4 | **37.2±2.6** |
| | **WeightCLIP**-MBM | 61.2±1.9 | **84.7±1.5** | **55.2±3.4** | 53.1±2.9 | 67.2±3.7 | 35.6±1.0 |
| 10 | tr. fr. scratch | 71.4±2.3 | 91.1±0.9 | 56.7±9.2 | 64.1±0.4 | **95.0±1.8** | 50.3±1.7 |
| | TANS | 76.7±0.8 | 91.9±1.4 | 76.9±4.2 | 62.6±1.2 | 94.9±0.8 | 56.2±3.1 |
| | **WeightCLIP**-LM | **79.0±0.3** | 92.3±0.9 | 82.2±1.5 | **64.5±0.4** | 94.2±2.0 | **57.2±1.3** |
| | **WeightCLIP**-MBM | 78.3±0.4 | **93.0±1.0** | **86.5±2.3** | 64.3±0.8 | 93.1±1.2 | 54.7±0.4 |

*Table 9.* Sensitivity to dataset prompt size. OOD generation reports test accuracy after one epoch of fine-tuning, averaged over the OOD datasets from the main paper. Dataset cls. reports dataset-ID classification accuracy of the DeepSets encoder. All values are percentages.

| Prompt size | OOD gen. (%) | Dataset cls. (%) |
|---|---|---|
| 1 image | 61.1±6.4 | 94.3 |
| 10 images | 62.7±4.4 | 97.5 |
| 32 images | **64.3±1.2** | 99.2 |
| 100 images | 63.8±1.7 | **99.8** |

### B.3. Sensitivity analysis on the subset size

We ablate the dataset prompt sizes of 1, 10, 32, and 100 images using 5 random prompts per setting. We evaluate both OOD weight generation after one epoch of fine-tuning, and standalone dataset classification accuracy, where the DeepSets encoder is trained to predict the dataset identity from the sampled image set. As shown in Tab. 9, both metrics are already strong with a single image and change only modestly with larger prompts. This suggests that the DeepSets encoder captures dataset-level signal even from very small prompts, and that the downstream performance is not strongly dependent on the particular sampled subset.

## C. Implementation details

This section summarizes the experimental configuration needed to reproduce the implementation of the approach, detailing datasets, architectures, and optimization settings.

### C.1. Datasets

We summarize the datasets used to construct the model zoos described in Section 4.2 in Tab. 10. We summarize the datasets used to construct the model zoos described in Section 4.2 in Tab. 10. The datasets are drawn from the TANS (Jeong et al., 2021) meta-dataset collection, but restricted to 20 training datasets for the CNN zoo and 10 for the ResNet18 zoo. As in TANS, we cap datasets to at most 20 classes during zoo construction. When available, we cite the original dataset papers for the Kaggle sources, including Aerial Cactus (López-Jiménez et al., 2019), BreakHis (Spanhol et al., 2016), NumtaDB (Alam et al., 2018), CT Images (Hssayeni, 2020; Hssayeni et al., 2020), EuroSAT/Land Use (Helber et al., 2019), Colorectal Histology (Kather et al., 2016), COVID-19 Radiography (Chowdhury et al., 2020; Rahman et al., 2021), Honeybee Pollen (Rodriguez et al., 2018), and CIFAR-10 (Krizhevsky & Hinton, 2009).

### C.2. Model Zoos

Each dataset contributes 500 or 100 independently trained models (different random seeds), yielding 10,000 CNN models and 1,000 ResNet18 models in total. Tab. 11 summarizes the training configuration for the zoo models.

*Table 10.* Datasets used for model-zoo construction and OOD evaluation. "Instances" are reported as train/val counts.

*(a)* Train datasets used to construct the CNN model zoo

| No. | Dataset Name | Instances | Cls. |
|---|---|---|---|
| 1 | Traffic Signs | 5735 / 717 | 8 |
| 2 | Aerial Cactus | 17199 / 2150 | 2 |
| 3 | Big Cats | 2875 / 360 | 4 |
| 4 | Four Shapes | 11976 / 1496 | 4 |
| 5 | Bottles | 11992 / 1499 | 5 |
| 6 | Car Models | 3229 / 405 | 45 |
| 7 | Simpsons | 15969 / 1999 | 39 |
| 8 | Simpsons 3 | 16709 / 2090 | 39 |
| 9 | Breast Cancer Tissues | 6323 / 789 | 8 |
| 10 | Mushrooms | 5312 / 662 | 9 |
| 11 | Artworks | 6997 / 877 | 51 |
| 12 | Bengali Digits | 57620 / 7203 | 10 |
| 13 | Lego Bricks | 5103 / 638 | 16 |
| 14 | Deepfake Detection | 12000 / 1500 | 2 |
| 15 | Furniture | 5186 / 648 | 5 |
| 16 | CT Images | 4255 / 532 | 2 |
| 17 | Land Use | 14399 / 1801 | 10 |
| 18 | Casting | 6866 / 858 | 2 |
| 19 | Manual Alphabet | 69600 / 8700 | 29 |
| 20 | Cassava Leaf | 17115 / 2141 | 5 |

*(b)* Train datasets used to construct the ResNet18 model zoo

| No. | Dataset Name | Instances | Cls. |
|---|---|---|---|
| 1 | Artworks | 6997 / 877 | 51 |
| 2 | Blood Cells | 9954 / 1244 | 4 |
| 3 | Breast Cancer Tissues | 6323 / 789 | 8 |
| 4 | Aerial Cactus | 17199 / 2150 | 2 |
| 5 | Cassava Leaf | 17115 / 2141 | 5 |
| 6 | CT Images | 4255 / 532 | 2 |
| 7 | Land Use | 14399 / 1801 | 10 |
| 8 | Lego Bricks | 5103 / 638 | 16 |
| 9 | Real/Fake Legos | 36606 / 4576 | 4 |
| 10 | Casting | 6866 / 858 | 2 |

*(c)* Test datasets used for OOD evaluation

| No. | Dataset Name | Instances | Cls. |
|---|---|---|---|
| 1 | Colorectal Histology | 4000 / 496 | 8 |
| 2 | COVID-19 | 2298 / 288 | 3 |
| 3 | Speed Limit Signs | 272 / 35 | 4 |
| 4 | Honeybee Pollen | 571 / 71 | 2 |
| 5 | Real or Drawing | 4000 / 500 | 10 |
| 6 | CIFAR-10 | 45000 / 5000 | 10 |

**Architectures.** The CNN is designed for $32 \times 32$ RGB images. It consists of three convolutional blocks followed by two fully connected layers. GELU activations are used throughout to improve robustness across heterogeneous datasets. Resulting in approximately 11K parameters.

The ResNet-18 models are a modified version using a channel width multiplier of 0.5. Thus channel dimensions are reduced to $\{32, 64, 128, 256\}$, resulting in 2.8M parameters. This choice was made because of compute constraints.

*Table 11.* Model zoo training hyperparameters.

| Hyperparameter | CNN | ResNet18 |
|---|---|---|
| Optimizer | AdamW | SGD (momentum) |
| Initialization | Normal | Normal |
| Act. Function | GELU | ReLU |
| Learning Rate | Sweep | Sweep |
| Weight Decay | $3 \times 10^{-3}$ | $5 \times 10^{-4}$ |
| Dropout | – | 0.15 |
| Batch Size | 128 | 256 |
| Epochs | 45 | 45 |
| Scheduler | OneCycleLR | OneCycleLR |
| Gradient Clipping | 1.0 | – |
| Models / Dataset | 500 | 100 |

### C.3. Weight Space Autoencoder

Following SANE (Schürholt et al., 2024) we represent each network as a sequence of tokens obtained from the flattened parameter vector. Tokens are grouped into windows and processed by a transformer encoder–decoder. Tab. 12 reports the exact transformer configuration. For the generation experiments we use 100 subsamples and report the top-$m$ result with $m = 5$. Additional training parameters are provided in the released code.

*Table 12.* The autoencoder configurations and training epochs for CNN and ResNet18 experiments.

| Parameter | CNN | ResNet18 |
|---|---|---|
| Token Size | 201 | 288 |
| Latent Dimension | 128 | 192 |
| $d_{\mathrm{model}}$ | 1024 | 1600 |
| Layers | 16 | 12 |
| Attention Heads | 16 | 20 |
| Block Size | 118 | 512 |
| Dropout | 0.10 | 0.05 |

### C.4. Dataset encoder (DeepSets)

DeepSets(Zaheer et al., 2017) encodes small sets of images into a permutation-invariant dataset embedding. As described in the main paper we train the encoder with an auxiliary dataset-classification objective, the weight of this objective is initialized to 1.0 and annealed over the training run. Tab. 13 reports the training hyperparameters of the dataset encoder.

*Table 13.* DeepSets dataset encoder training configuration.

| Parameter | Value |
|---|---|
| Image Size | 32×32 |
| Set Size | 10 |
| Embedding Dim | 128/192 |
| Aggregation | Sum |
| Classification Weight Init. | 1.0 |
| Classification Annealing Epochs | 600 |

### C.5. Dataset–model alignment loss

We use a SigLIP-style bidirectional contrastive loss between dataset embeddings and model embeddings, with a learnable logit bias. The loss is applied directly in the autoencoder latent space, so alignment shapes the representations used later for retrieval, mapping, and refinement rather than only a separate projection head. The hyperparameters are shown in Tab. 14.

*Table 14.* Dataset–model alignment loss configuration.

| Component | Value |
|---|---|
| Temperature | 1.0 |
| Alignment Weight | 0.25 |
| Logit Bias Init | -4.0 |

### C.6. Memory bank mapper

The main paper describes the memory-bank mapper as a convex combination over stored model tokens. Here we describe how the mapper is trained and how it is instantiated for the CNN and ResNet18 zoos. During training, the memory bank stores the token sequences of all training models and is kept fixed. Instead of supervising the mapper with one-hot targets, we use a soft neighborhood-based target that reflects similarity in model-latent space.

Let $\bar{z}^{(m)} = \frac{1}{L} \sum_t z_t^{(m)}$ be a pooled representation of model $m$ and define

$$d(i,j) = \|\bar{z}^{(i)} - \bar{z}^{(j)}\|_2. \tag{12}$$

Let $\mathrm{TopK}(i)$ denote the indices of the $K$ nearest neighbors of model $i$ under this distance. For a training example $i$, we define a soft target distribution over its neighbors as

$$q_i(j) \propto \exp(-d(i,j)) \quad \text{for } j \in \mathrm{TopK}(i), \tag{13}$$

The mapper predicts a distribution $p_{i,t}$ over memory-bank entries for each token position $t$, and is trained with a soft cross-entropy loss averaged over token positions:

$$\mathcal{L}_{\mathrm{MB}} = -\frac{1}{BL} \sum_{i,t} \sum_{m=1}^{N} q_i(m) \log p_{i,t}(m). \tag{14}$$

The implementation differs between ResNet18 and CNN models in how the logits are parameterized. For ResNet18, the longer token sequence is handled with a position-conditioned MLP: the dataset embedding is broadcast over token positions, combined with learned position prototypes, and mapped to memory-bank logits by a shared MLP. For the smaller CNN models we can use a transformer-based mapper with the dataset embedding as a conditioning token and learned query tokens for each model-token position. Both variants optimize the same objective and use the same memory-bank expectation at inference optionally with temperature scaling and top-$K$ filtering before decoding.

## C.7. TANS (Task-Adaptive Neural Network Search)

We adapt TANS (Jeong et al., 2021) to our uniform-architecture model zoos. Since all models within each zoo share the same architecture, the topology encoding used in the original OFA-based TANS setup is uninformative and is removed. Each model is instead represented only by the functional embedding, computed by passing a fixed bank of Gaussian noise images through the model and flattening the penultimate-layer activations.

The query encoder follows TANS: a small set of target-dataset images is embedded with an ImageNet-pretrained ResNet18, projected, mean-pooled, and normalized. We train the dataset–model retrieval space with the original bidirectional hard-negative contrastive loss and accuracy-prediction objective. At test time, we retrieve models by dot-product similarity and re-rank the top candidates using the predictor. Retrieved models are then evaluated directly or fine-tuned with the same protocol as our other baselines.

## C.8. Hypernetwork (Text2Model)

We include a hypernetwork baseline adapted from Text2Model (Amosy et al., 2024). Since our setting does not use textual task descriptions, we replace the text encoder with our DeepSets dataset encoder, which maps a small image set from the target dataset to a single dataset embedding. In contrast to the original Text2Model setup, which generates only a classifier head on top of a frozen backbone, our baseline generates the full parameters of ResNet18 model.

To make full-model generation tractable we use the same tokenization as in our main approach. The hypernetwork predicts a sequence of weight tokens, which are then deto-kenized into a complete ResNet18 state dictionary. Architecturally, the dataset embedding is first processed by Text2Model-style EVLayer blocks, broadcast across state-dict token positions, and combined with learned positional embeddings. A shared per-token decoder then maps each position-conditioned feature to a weight token.

The baseline is trained with the Text2Model meta-learning objective. For each training dataset, the hypernetwork generates an initial model, which is adapted by a small number of inner-loop SGD steps. The displacement between the generated and adapted weights is used as a surrogate gradient for updating the hypernetwork, avoiding explicit backpropagation through the full inner loop. Since Text2Model relies on pretrained text features, we warm-start our dataset encoder using the classification loss before hypernetwork training. At test time, an unseen dataset prompt is encoded, mapped to a full ResNet18 state dictionary, and evaluated directly or after the same fine-tuning protocol as the other baselines.

## D. Additional geometry diagnostics

To further analyze the effect of dataset–model alignment, we look at the angular structure of the token embeddings via cosine distances. These diagnostics allows us to assess whether alignment induces meaningful dataset structure.

### D.1. Pairwise cosine distances

We compare intra-dataset and inter-dataset token similarities under training without alignment and with alignment. The results show that alignment produces a clear separation between intra-dataset and inter-dataset cosine distances, while training without it yields largely overlapping similarity distributions.

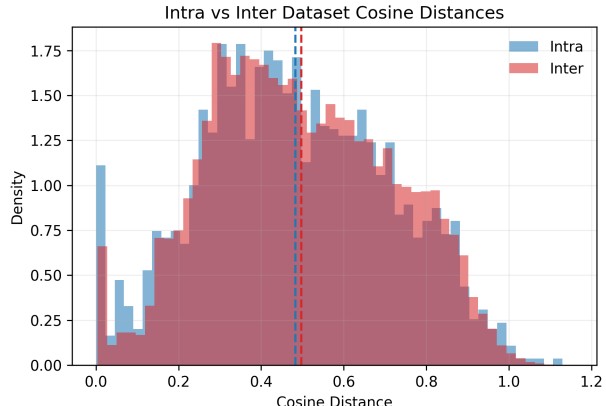

*(a)* Pairwise cosine distances between token embeddings trained without alignment loss. Intra-dataset and inter-dataset distances largely overlap, indicating little dataset-level angular structure.

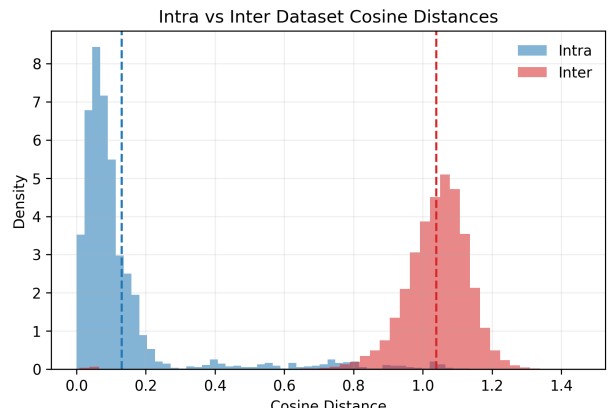

*(b)* Pairwise cosine distances between token embeddings after using the alignment loss. Intra-dataset distances are consistently smaller than inter-dataset distances, showing that alignment organizes token directions by dataset.

*Figure 7.* Pairwise cosine distance distributions between token embeddings with and without dataset–model alignment. Red curves denote intra-dataset distances and blue curves denote inter-dataset distances. Alignment induces a clear separation between datasets, whereas training without it yields overlapping distributions.

## D.2. Alignment matrix

We further analyze the global structure induced by alignment using a model–dataset similarity matrix in Fig. 8. Each entry measures the cosine similarity between an averaged model embedding and a dataset embedding in the aligned latent space. The resulting matrix exhibits a clear diagonal structure, indicating that alignment consistently associates models with the correct dataset semantics. Off-diagonal similarities are generally low, with the main exceptions occuring between the *Simpsons Characters* and *Simpsons Challenge* datasets. Two examples which share highly overlapping visual content and class structure. This residual confusion is therefore expected and reflects genuine dataset similarity rather than a misalignment of model and dataset representations. The matrix uses the original Kaggle names, which may differ from the shortened names used elsewhere in the paper.

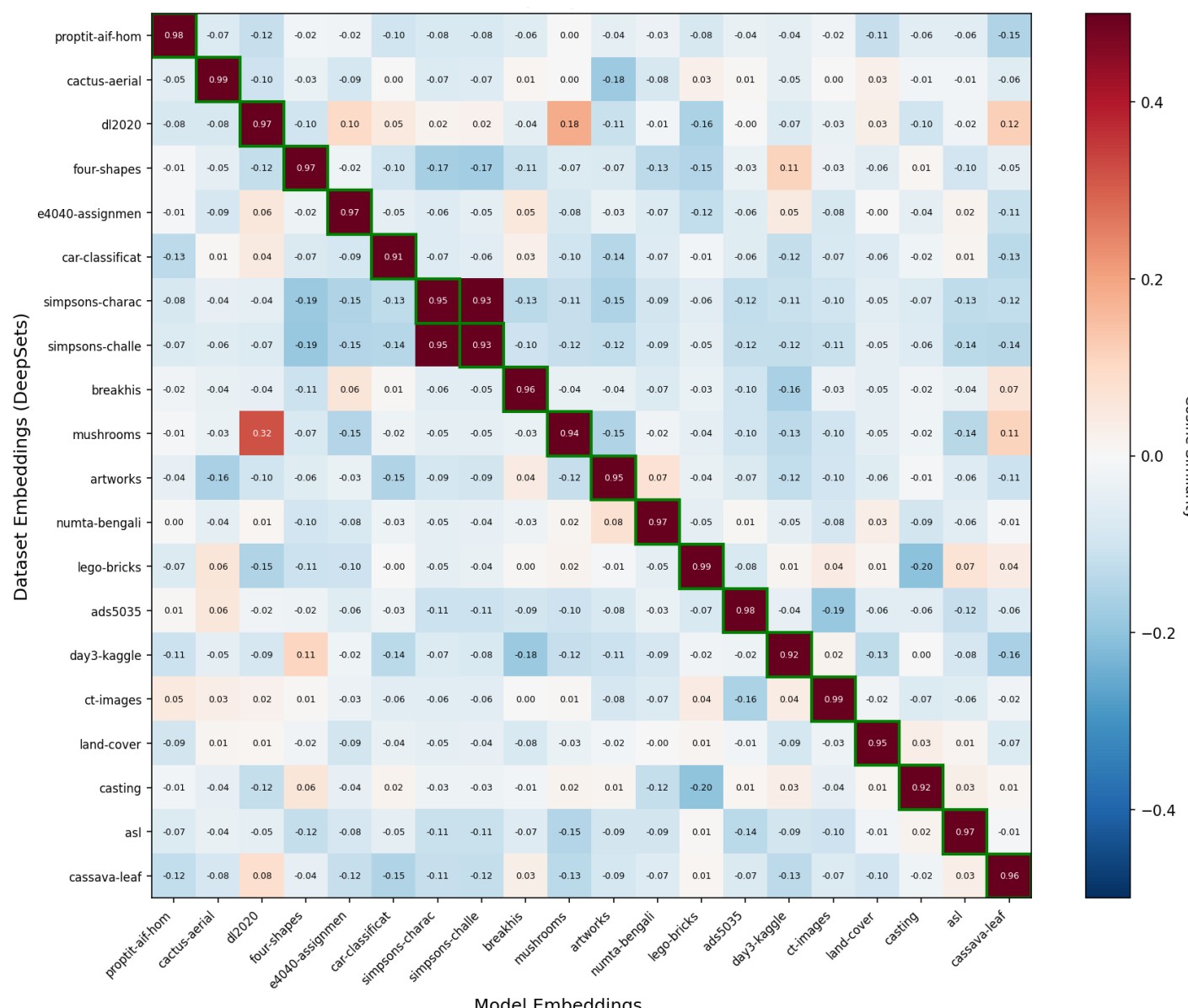

*Figure 8.* Model–dataset cosine similarity matrix induced by the aligned latent space after triangulation. Rows correspond to dataset embeddings and columns to the token embeddings. Strong diagonal structure indicates consistent alignment between models and their originating datasets, with limited off-diagonal confusion primarily between semantically overlapping datasets.

