# OpenReview forum: "WeightCLIP: Aligning Datasets and Models for Weight Space Learning"
_ICML.cc/2026/Conference — ICML 2026 regular_

### Official Review · Reviewer_yLjV · 2026-03-05

**Soundness:** 2
**Presentation:** 3
**Significance:** 2
**Originality:** 2
**Overall Recommendation:** 4
**Confidence:** 3

**Summary:**

The paper studies learned weight space representation. It proposes a method to align weight representations with the datasets used to train the corresponding models by learning a shared latent space. The approach combines a weight-space autoencoder that encodes model parameters with a dataset encoder that embeds dataset samples. The two representations are aligned through a contrastive objective to encourage models trained on similar datasets to lie close together in the latent space. Experiments show that incorporating dataset information improves retrieval, model generation, and test-time refinement compared to standard weight-space learning approaches.

**Compliance With Llm Reviewing Policy:**

Affirmed.

**Final Justification:**

After reading the reviewers' comments and the authors' response, I believe my concerns are partially addressed. Including (an extended version of) the additional provided results and details will strengthen the submission. Hence, I update my score accordingly (3 $\to$ 4). I encourage the authors to revise the paper to include these new results.

**Key Questions For Authors:**

Please see strengths and weaknesses section.

**Limitations:**

Yes.

**Strengths And Weaknesses:**

**Strengths:**
- The paper is generally well structured.
- The motivation for a better-structured latent space is clear.
- The experimental section is well designed, with a good ablation study.


**Weaknesses:**
- The method includes several heuristic design choices (e.g., memory-bank token mixing, spherical prior projection, data embedding to weights projection, etc.), which make the approach somewhat complex and potentially sensitive to hyperparameters, model/data representative choices, etc.
- The motivation of providing structure to the latent space is clear, but it is not clear why aligning representation with the dataset directly should work better than other alternatives, like aligning models trained using the same data.
- Following on that, the task of learning a joint space is somewhat niche. One of the main motivations is better model generation, but there are not enough direct comparisons with strong SoTA weights generation baselines.
- In general, the baselines are relatively narrow (one designed method and a retrieval-style baseline), and it is unclear how the method compares to stronger or more conventional alternatives. While this may partly stem from the niche problem formulation, the comparison remains quite limited. I suggest adding additional weights generation baselines based on weight space networks.
- Since the weight space encoder is not really used in the experiments, another alternative would be using hypernetwork-based baselines to directly map datasets to model weights, see e.g. [1].
- Also, can include natural contrastive learning based approaches, see questions below.
- The memory-bank mapper constrains generated representations to convex combinations of tokens from training models. While this may regularize generation, it also limits the ability to produce genuinely novel models and makes the method dependent on the coverage and composition of the memory bank.

Questions:
- The dataset encoder uses a subset of images. How sensitive is the training/inference process to the choice of this set?
- Experiments focus on a relatively narrow set of architectures and datasets.
- Since generation relies on convex combinations of training model representations, it is unclear to what extent the method produces genuinely novel models versus interpolations of existing ones. How can we be sure the generated models are indeed novel?
- Have you tried using a CL loss to encourage models trained with the same data to align in the latent space?
- Another approach is to encourage similarity between a model and a noisy version of it, like prior works, e.g. [2].

[1] Amosy et. al., Text2Model: Text-based Model Induction for Zero-shot Image Classification.

[2] Shamsian et. al., Improved Generalization of Weight Space Networks via Augmentations.

---

> ### Author Rebuttal · Authors · 2026-03-30
>
> We thank Reviewer yLjV for their review as we respond to the reviewer’s questions and concerns and provide additional experiments.
> ## Subset sensitivity
>
> We agree that this is important, as our goal is for the dataset encoder to capture dataset-level structure rather than prompt-specific artifacts. We therefore ablated subset sizes of **1 / 10 / 32 / 100** images, and compared the DeepSets encoder used in the paper to a **SetTransformer** alternative. Both were evaluated in (i) OOD weight generation and (ii) standalone **dataset classification**. Results are reported for the 3 OOD datasets in the main paper after 1 epoch of finetuning for weight generation results, using 5 random dataset prompts each time.
>
> For DeepSets, OOD generation was **61.1±6.4 / 62.7±4.4 / 64.3±1.2 / 63.8±1.7**, showing strong performance even with 1 image and only modest changes with subset size. The dataset classification resulted in **94.3 / 97.5 / 99.2 / 99.8** for the same subset sizes. So the DeepSets encoder is already capable of learning a discriminative dataset representation from small sets. Thus, the weak dependence on prompt size is already visible in the dataset encoder, not only in downstream components.
>
> Because it may seem surprising that 1 image is sufficient, we ran a **sanity check**. We randomly split a dataset into 10 subsets, treated them as different “datasets” and trained the dataset encoder to classify them. Performance is then at chance, as expected. This shows that, in our setting, 1 image indeed contains enough signal to distinguish genuinely different datasets.
>
> Repeating the same experiments with a SetTransformer encoder gave **55.1±3.7 / 65.6±3.2 / 63.8±1.3 / 63.6±1.8** for the subset sizes in question. Here prompt size matters more, with performance increasing for bigger sizes. It shows that prompt-size sensitivity depends on the encoder design, with the DeepSets encoder used in the paper remaining robust even for very small prompts.
>
> We also explored auxiliary choices. Using an ImageNet-pretrained feature extractor inside the dataset encoder did not help OOD generation, **53.2% vs. 64.2%**, suggesting that generic pretrained features are not necessarily the right representation for distinguishing datasets in our setting. Likewise, removing the auxiliary dataset-classification loss is clearly harmful, **43.4% vs. 62.2%**.
>
> These experiments show that the dataset prompt **is not sensitive** in the way implied by the review. DeepSets is robust with very small prompts, the sanity check confirms that this is not due to trivial artifacts, and the low variance across random subsets further shows that the method is not strongly dependent on a particular sampled prompt.
>
> ---
> ## Narrow scope
>
> To address this concern we extended the experiments to a mixed-architecture zoo containing **ConvNeXt, MobileNetV2, and ResNet18**, using an architecture-specific mapper. After 1 epoch on OOD datasets, the average accuracies are **57.0 / 58.1 / 65.6** respectively, compared to 41.7/ 42.4 / 40.3 for tr. fr. scratch. These runs used a shorter training budget than the main paper due to compute constraints. While this does not fully explore the heterogeneous-zoo problem it shows that the core alignment idea is not inherently tied to a single architecture family.
>
> ---
> ## Novelty of generated models
>
> 2 points are important here. First, the memory-bank mapper is only 1 of all proposed mapping mechanisms: we also evaluate a simple linear mapper, which is not restricted to convex combinations of training latents and already outperforms retrieval and the D2NWG baseline. Second, even for the memory-bank mapper, the relevant question is whether it goes beyond nearest-neighbor selection.
>
> We report retrieval results in the App. Table 8 and add new results for ResNets for the 3 OOD datasets with acc of **25.6 / 59.4 / 81.3** for epochs 0/1/10 respectively. These results are lower than our generated models with acc of **26.2 / 75.3 / 88.4**. Since the generated models outperform picking the best available model, **they must represent different weights**.
>
> ---
> ## Why align with the dataset directly
>
> This is a central point of this work. Aligning models trained on the same dataset could encourage dataset-conditioned clustering in latent space, but it would not provide a dataset representation that can be used at test time to prompt the model latent space. Our goal is therefore not just to cluster models by dataset identity, but to learn a **shared cross-modal latent space** where dataset prompts can be used for weight generation and subsequent latent refinement. For the same reason, objectives that enforce similarity to noisy versions of the same model, are interesting but are better understood as regularizers and cannot establish the image-to-model bridge central to our approach.
>
> ## Baselines and hypernetwork comparisons
>
> Due to space limitation we kindly refer to our answer for reviewer MHWa, where we evaluate against **Text2Model**.

---

> > ### Author Rebuttal · Reviewer_yLjV · 2026-04-02
> >
> > I thank the authors for their response and for providing additional details and results.
> > The response partially addresses my concerns.
> > - Regarding the generated model's novelty: "Since the generated models outperform picking the best available model, they must represent different weights." -- Do the reported results correspond to (oracle choice) of best available models or a retrieval baseline? I believe a more thorough analysis is needed to support such a decisive statement.
> > - Regarding the narrow experimental scope and limited baselines: I thank the authors for adding an HN-based baseline and providing initial results with a mixed-architectures zoo. I encourage the authors to extend the baseline coverage further and provide wider results on mixed arch. in the revised version of the manuscript.
> >
> > That said, I do believe adding these results to the manuscript strengthens the submission, and so I raise my score.

---

> > > ### Author Response · Authors · 2026-04-07
> > >
> > > Thank you for the follow up and for raising your score after considering our rebuttal and additional results. We appreciate your positive assessment that these additions strengthen the submission.
> > >
> > > Regarding the novelty point, the comparison we intended for is against the retrieval baseline rather than an oracle selection over available models (because of practicality, see our rebuttal to Reviewer r5of). We agree that our original phrasing was too strong, and we will revise this statement in the paper to make the comparison precise and avoid overstating the conclusion. We will also include the additional mixed-architecture and HN results into the revised paper to better address the concerns about evaluation scope.

---

### Official Review · Reviewer_9Vhw · 2026-03-11

**Soundness:** 3
**Presentation:** 3
**Significance:** 3
**Originality:** 3
**Overall Recommendation:** 4
**Confidence:** 1

**Summary:**

The paper introduces a dataset-aligned latent space for neural network weights. A weight autoencoder first encodes model parameters into latent tokens, while a dataset encoder maps a small subset of target samples into a dataset embedding. A bidirectional contrastive loss is then used to align these two spaces, with the goal of making the weight latent space semantically structured by the underlying training data. Based on this aligned space, the method learns a mapper from dataset embeddings to model latent tokens, which are subsequently decoded into network weights. The generated latent can also be further optimized under a hyperspherical-shell prior via task-loss backpropagation through the decoder. The main claimed advantages are improved dataset-conditioned model retrieval, stronger weight generation from data prompts, and more effective latent refinement compared with conventional fine-tuning.

**Compliance With Llm Reviewing Policy:**

Affirmed.

**Key Questions For Authors:**

See questions above.

**Limitations:**

See limitations above.

**Strengths And Weaknesses:**

# Strengths:
1. A key motivation of the paper is that existing weight-space latent representations are often not semantically organized by the underlying training data. This is an important and well-motivated limitation, and the paper’s attempt to use datasets as a semantic reference frame is both intuitive and interesting.
2. Aligning dataset representations with model-weight latents is a simple but effective idea. It gives the method a clean conceptual story. Instead of treating the dataset only as side information, the method explicitly reshapes the latent space so that dataset structure is reflected in the geometry of model representations.
3. A strength of the paper is that the same aligned latent space is used for several downstream tasks, including dataset-to-model retrieval, dataset-conditioned weight generation, and latent refinement. This makes the contribution feel more coherent and substantial than a method designed for only one isolated application.
4. The reported results suggest that the proposed alignment improves retrieval and generation performance, and the refinement experiments further indicate that the learned latent space is useful beyond simple reconstruction.

# Weaknesses:
1. The final performance may depend substantially on the mapper design, especially the memory-bank variant, rather than solely on the aligned latent space itself.

---

> ### Author Rebuttal · Authors · 2026-03-30
>
> We thank Reviewer 9Vhw for their positive review and for highlighting an important question: whether the final gains mainly come from the mapper design, especially the memory-bank mapper, rather than from the aligned latent space itself.
>
> We believe that **Table 3** particularly shows that when applying the same type of mapper across different settings the **strongest gains appear only when the latent spaces are aligned**. This directly supports our main claim that the benefit comes from **restructuring the latent space**, not merely from adding a mapper on top.
>
> We also note that the effect **is not specific to the memory-bank mapper**. Besides the memory-bank variant, we evaluate a simple linear mapper, which is not restricted to convex combinations of training latents. This linear mapper already **outperforms both retrieval and the D2NWG baseline**.
>
> We hope this makes this point clearer, and we will revise the paper to emphasize more strongly that: **(i)** alignment improves performance even with simple mappers, and **(ii)** the gains in Table 3 arise specifically when the weight latent itself is reshaped by dataset alignment.

---

> > ### Author Rebuttal · Reviewer_9Vhw · 2026-04-03
> >
> > Thanks for your rebuttal. You have a good clarification for me who are not familiar with this field.
> >
> > I will keep my score.

---

> > > ### Author Response · Authors · 2026-04-07
> > >
> > > Thank you for your followup, we are glad that our clarification addressed your concern and that you now consider the issue fully resolved.

---

### Official Review · Reviewer_r5of · 2026-03-11

**Soundness:** 2
**Presentation:** 3
**Significance:** 2
**Originality:** 3
**Overall Recommendation:** 4
**Confidence:** 4

**Summary:**

This paper aims to learn a shared embedding space for model weights and data. Essentially, it uses a contrastive objective, but the approach has may fine details. It is validated on datasets that the authors created.

**Compliance With Llm Reviewing Policy:**

Affirmed.

**Final Justification:**

The authors analysed the difference from previous work and promised to add this discussion to the intro. They provided clarifications on the experimental protocol that make it appear more solid than I initially thought. I have revised up my score.

**Key Questions For Authors:**

Can you comment on the difference from prior work?
Why are you not taking weight symmetries into account?
Can you provide more justification for the experimental. protocol?

**Limitations:**

The above issues are not mentioned as limitations.

**Strengths And Weaknesses:**

Weight space learning, developing machie learning for weights, is a very interesting, timely and important task. In particular, developing multimodal weight space approaches is interesting and potentially useful. The paper is also well presented.

However, this paper has critical weaknesses.

First, mapping models and datasets to the same space is not new e.g., Learning on LORAs [A] or ProbeX [B]. Both papers predict the training dataset given the weights, i some cases also in a zero-shot manner (by mapping to clip). This paper states that its first contribution is "Dataset-aligned Latent Space". As there is at least some previous work on this, it would be useful to explain the difference from this prior work, and also compare to it.

Second, the architecture is quite ad-hoc and fails to account for the parameter symmetries of model weights which are known to be critical for developing effective approaches. This was done in works [A,B]. Given that this work did not do it, there are two possible hypothesis: (i) that it is not needed (ii) that it might improve results. It would be instructive to esbalish which of these hypotheses is correct.

Finally, it is not clear the experimental protocol is sound. For example, the existence of different checkpoint from the same run in train and test may lead to serious leakage.  Also the OOD performance is not much better than random. Another issue is that model generation is not compared to picking the best available model, which is a much easier task.

[A] Putterman, Theo, Derek Lim, Yoav Gelberg, Stefanie Jegelka, and Haggai Maron. "Learning on LoRAs: GL-Equivariant Processing of Low-Rank Weight Spaces for Large Finetuned Models." Workshop on Neural Network Weights as a New Data Modality, ICLR'25.
[B] Horwitz, Eliahu, Bar Cavia, Jonathan Kahana, and Yedid Hoshen. "Learning on model weights using tree experts." CVPR'25

---

> ### Author Rebuttal · Authors · 2026-03-30
>
> We thank Reviewer r5of for their review and would like to politely correct several arguments of this review, which we believe are misunderstandings of our submission.
>
> ---
> ---
> ## Prior work
>
> Reviewer r5of cites [A, B] as prior works that already “map[...] models and datasets to the same space”. We discuss their relation to our work hereinafter. We agree these papers are related, especially [B], and will discuss them more explicitly in the revision. However, neither addresses our weight generating task setting directly.
>
> [A] trains models to predict properties from the weights of LoRA adapters. While it shows it is possible to predict the fine-tuning dataset from its weights successfully, this paper does not address the inverse transformation going from dataset space to model space to generate weights as we do.
>
> [B] proposes a probing-based method to learn from the weights of a single hidden model layer. [B] predicts properties of the training dataset but also aligns weights-to-text representations. From this experiment setup, it performs several downstream tasks, such as zero-shot model classification, model retrieval and one-class classification.
>
> While there are indeed similarities between both approaches, we believe that our work is different to ProbeX’s wrt:
> 1. It (ProbeX) studies fine-tunes trained on <10 samples from a single class, whereas our method works on encoding different datasets as a whole.
> 2. It relies on model-tree experts, with the need to construct one for every single tree, whereas our approach is independent of such lineage structures.
> 3. It aligns weight embeddings with text embeddings whereas we encode the dataset and therefore do not need a textual description.
> 4. It performs model retrieval, not weight generation or latent-space refinement of the learned representation.
> 5. As mentioned by the authors of ProbeX, OOD generalisation of ProbeX is limited, whereas we show promising model retrieval performance for OOD datasets as well as for generation.
>
> We do agree that the papers mentioned by Reviewer r5of are relevant to our work, and will discuss it in our introduction, but nevertheless remain convinced that our submission presents novel and impactful methods to align dataset and weight-space representations for weight generation.
>
> ---
> ---
> ## Symmetries
>
> The reviewer is right in stressing that symmetry-equi/invariant encoders have been successful in weight-space learning. However, most such works focus on discriminative tasks, model editing or learning to optimize, whereas recent generative methods do not use symmetry-aware architectures. Our method follows this latter line where **symmetries are learned using augmentations**. Making the weight encoder symmetry-invariant would be an interesting extension to potentially further improve the weight generation results.
>
> ---
> ---
> ## Experimental setup
>
> Reviewer r5of mentions a potential leakage issue, suggesting that checkpoints from the same run may appear in both train and test. This appears to be a misunderstanding. As stated in Section 4.2, we reserve six downstream datasets exclusively for testing, and for these datasets there are **no corresponding model zoos at all**: they are only used as image datasets for the OOD prompts. Following, checkpoints from the same run cannot be present in both training and evaluation. We will clarify this point explicitly in the revised manuscript
>
> ---
> ---
> ## OOD performance
>
> Reviewer r5of states that the OOD performance is “not much better than random”. We respectfully disagree. In the OOD generation setup, none of the generated models has been trained on the target dataset by definition, so low zero-shot absolute accuracies are expected, yet we still outperform both baselines. As seen in Table 2 we are **+7.6%** above random for zero shot, with this number increasing to **+35%** after 1 epoch of finetuning (compared to tr. fr. scratch). This is expected because (i) classifier-head size mismatches make pure zero-shot decoding harder, and (ii) weight generation eval benchmarks are typically evaluated as initializations with subsequent fine-tuning.
>
> ---
> ---
> ## Picking the best available model
>
> We interpret this as referring to an oracle selection baseline that exhaustively evaluates every model in the train zoo on the target OOD dataset (where there is no model zoo) and then picks the best one. While this could be a useful upper bound it is not the practical setting targeted by this work, as it requires expensive brute-force evaluation of the full zoo on the new task.
>
> The practical non-generative baseline in our setting is retrieval. We report retrieval results in the appendix (Table 8) and add new results for ResNets for the 3 OOD datasets from Sec 4.5 with acc. of **25.6 / 59.4 / 81.3** for epochs 0/1/10 respectively. These results are lower than our generated models with acc. of **26.2 / 75.3 / 88.4**. Generated models perform stronger than picking the best available model in a practical setup.

---

> > ### Author Rebuttal · Reviewer_r5of · 2026-04-03
> >
> > I have read the author responses and the other review. I thank the authors for provided the analysis re-previous work, and clarification on the experimental protocol, which is better than I initially thought. Not considering equivariance directly is regrettable but perhaps not to the point of blocking publication. I will raise my score to 4.

---

> > > ### Author Response · Authors · 2026-04-07
> > >
> > > Thank you for the followup, reading both our rebuttal and the other reviews and updating your score. We are glad the clarifications helped address your concerns. We also appreciate your comment regarding equivariance and we agree this is an interesting direction.

---

### Official Review · Reviewer_MHWa · 2026-03-15

**Soundness:** 3
**Presentation:** 3
**Significance:** 3
**Originality:** 3
**Overall Recommendation:** 5
**Confidence:** 4

**Summary:**

This paper studies weight space learning (WSL) with the goal of making learned model representations more semantically organized around the datasets that produced them. The core idea is to align a weight-space autoencoder latent with dataset embeddings produced by a DeepSets encoder, using a bidirectional contrastive loss. Rather than using dataset-model alignment only as a retrieval or conditioning interface, the paper uses alignment to reshape the geometry of the model latent space itself.

The paper introduces two downstream mechanisms. First, it learns dataset-to-model mapping modules that map a dataset embedding to a sequence of model tokens that can be decoded into network weights. Second, it proposes a latent refinement procedure that projects mapped latents onto a hyperspherical shell and then optimizes them through the decoder using target-task loss.

The paper evaluates the resulting representation on retrieval, generation/initialization, and refinement tasks across CNN and ResNet-18 model zoos, with results suggesting that alignment improves dataset-conditioned retrieval, produces stronger initializations for unseen datasets, and enables latent-space refinement that is competitive with or better than standard fine-tuning under matched budgets.

**Compliance With Llm Reviewing Policy:**

Affirmed.

**Final Justification:**

The rebuttal addressed my main concerns. I have updated my score.

**Key Questions For Authors:**

1. How sensitive is the approach to the dataset prompt size and prompt composition? Since the dataset encoder only sees 10 images, it would be helpful to know whether the downstream mapping/refinement results are stable for different set sizes or different prompt samples.

2. Can you clarify the intended practical regime in which this pipeline is preferable to direct training on the target dataset? A simple cost or amortization discussion would help. If the method is meant primarily for repeated transfer across many target datasets, stating that more explicitly would improve the paper.

**Limitations:**

Yes

**Strengths And Weaknesses:**

Strengths:
- The paper makes a conceptually clean contribution to WSL. The most interesting idea is to use alignment as a way to structure the weight latent space, rather than merely as a scoring or conditioning mechanism. This distinction is meaningful and novel.
- The method is modular and well designed. The paper decomposes the approach into alignment, mapping, and latent refinement, and each piece is evaluated in a reasonably interpretable way.
- The empirical package is strong within the stated scope. The paper does not rely on a single downstream metric; instead, it presents evidence from retrieval, model generation/initialization, refinement, interpolation, and latent geometry analyses.
- The dataset-to-model mapping results after short adaptation are strong. The one-epoch and ten-epoch results suggest that the method is practically useful as an initialization strategy, even if zero-shot decoded models are not the main success case.

Weaknesses:
- The evaluation scope is still narrow. All experiments are in image classification, all zoos are architecture-homogeneous, and the tested architectures are relatively small. This is acceptable for current WSL norms, but it limits how strongly one can generalize the conclusions.
- The practical value proposition would be stronger with a cost analysis. The pipeline assumes a pre-existing large model zoo plus training of the autoencoder, dataset encoder, and mapper. This may be worthwhile when amortized across many target datasets, but the paper does not quantify when that amortization becomes favorable relative to direct training.
- The paper’s generation story is best interpreted as dataset-conditioned initialization rather than strong zero-shot model synthesis. I do not view this as a fatal flaw, since the paper’s own experiments already emphasize the adapted setting, but the conclusion occasionally reads more broadly than the evidence supports.
- There are a few missing discussions in the broader landscape. Hypernetworks are relevant because they also map conditioning inputs to weights, with the main distinction here being that this paper uses a structured aligned latent space rather than a direct input-to-weights mapping. Likewise, model-merging/stitching or weight-space navigation literature would help position the work more fully. These are not critical omissions, but the related work section would benefit from them.

---

> ### Author Rebuttal · Authors · 2026-03-30
>
> We thank Reviewer MHWa for their review as we respond to the reviewer’s questions and concerns and provide **additional experiments**.
>
> ---
> ---
>  ## Dataset prompt sensitivity
>
> We agree that this is important, as our goal is for the dataset encoder to capture dataset-level structure rather than prompt-specific artifacts. We therefore ablated subset sizes of **1 / 10 / 32 / 100** images, and also compared the DeepSets encoder used in the paper against a **SetTransformer** alternative. Both were evaluated in **(i)** OOD weight generation and **(ii)** standalone **dataset classification**. Results are reported for the three OOD datasets in the main paper, after one epoch of finetuning for weight generation results. With 5 random dataset prompts used each time.
>
> For DeepSets the OOD generation resulted in **61.1±6.4 / 62.7±4.4 / 64.3±1.2 / 63.8±1.7**. Showing that it is already strong with a single image and changes only modestly with subset size. The standalone dataset classification resulted in **94.3 / 97.5 / 99.2 / 99.8** respectively for the same subset sizes. These results tell a similar story. The DeepSets encoder is already capable of learning a highly discriminative dataset representation from very small sets. Thus, the weak dependence on prompt size is already visible in the dataset encoder itself, not only in the downstream components.
>
> Because it may seem surprising that a single image is already sufficient, we ran a **sanity check**: we randomly split a dataset into 10 subsets and treated those subsets as if they were different “datasets,” and trained the dataset encoder to classify them. In this setting, performance is at chance level, as expected. This demonstrates that, in our setting, there is indeed **enough signal in a single image to distinguish genuinely different datasets**.
>
> We repeated the OOD generation experiments with a SetTransformer dataset encoder, resulting in **55.1±3.7 / 65.6±3.2 / 63.8±1.3 / 63.6±1.8** respectively for the subset sizes. Here prompt size matters more, with performance increasing for bigger sizes. It shows that prompt-size sensitivity depends on the encoder design, with the DeepSets encoder used in the paper remaining robust even for very small prompts.
>
> These experiments show that the dataset prompt **is not sensitive** in the way implied by the review. DeepSets performance is already robust with very small prompts, the sanity check confirms that this is not due to trivial artifacts, and the low variance across random subsets further shows that the method is not strongly dependent on a particular sampled prompt.
>
> ---
> ---
>  ## Cost analysis
>
> We agree that the intended practical regime should be stated more clearly. Our method is not meant to replace direct training in a single-task setting, but to operate in the **repeated-transfer regime**, where a learned aligned latent space is reused across many new target datasets (evaluated as OOD in our work). In that setting, the up-front cost is amortized over many models generated, with each new dataset only requiring prompt encoding, mapping, and short adaptation. We will make this regime explicit in the revision and add a brief amortization discussion.
>
> ---
> ---
>  ## Related work
>
> We agree with the reviewer that the paper should be positioned more explicitly relative to hypernetworks, as well as the broader literature on model-merging and stitching and will resolve this in the revised paper.
>
> As the reviewer correctly notes, hypernetworks are a relevant comparison, since they also map a conditioning signal directly to model weights. However our contribution is not merely another conditional generator, we learn a structured latent space over model weights whilst a hypernetwork only provides a direct input-to-weights mapping.
>
> To address this point directly we added a **hypernetwork based baseline**, besides the D2WNG baseline already reported, and evaluated it in the same generation setting as our method. We adapted a **Text2Model hypernetwork** to our dataset-conditioned setting: a DeepSets encoder maps an image subset to one dataset embedding, which conditions a Text2Model-style hypernetwork to generate full-model tokens for a ResNet18. Resulting in the average results (over three OOD datasets) **23.3 / 45.9 / 81.5** for 0/1/10 epochs, compared to our **26.2 / 75.3 / 88.4** results (Table 2).
>
> ---
> ---
>  ## Narrow evaluation scope
>
> To address this concern, we extended the experiments to a mixed-architecture zoo containing **ConvNeXt, MobileNetV2, and ResNet18**, using just an architecture-specific mapper. After 1 epoch on OOD datasets, the average accuracies are **57.0 / 58.1 / 65.6** respectively (compared to **41.7/42.4/40.3** for tr. fr. scratch). These runs used a shorter training budget than the main paper due to time constraints. While this does not fully explore the heterogeneous-zoo problem, it does show that the core alignment idea is not inherently tied to a single architecture family.

---

> > ### Author Rebuttal · Reviewer_MHWa · 2026-04-05
> >
> > The rebuttal addressed my main concerns. I have updated my score.

---

> > > ### Author Response · Authors · 2026-04-07
> > >
> > > We are pleased that we were able to address the concerns, thank you for the kind feedback and the score upgrade.

---

### Decision · Program_Chairs · 2026-04-30

**Decision:**

Accept (regular)

**Comment:**

This paper proposes to align dataset representations with neural network weight-space latents via a contrastive objective, with the goal of structuring the latent space and enabling downstream tasks such as retrieval, dataset-conditioned model generation, and latent refinement.

Reviewers agreed this is a timely and well-motivated contribution, and highlighted the conceptual clarity and novelty of reshaping the latent space via dataset–model alignment. The empirical evaluation, while somewhat narrow, is reasonably thorough and demonstrates consistent improvements, particularly for dataset-conditioned initialization and refinement.
Before rebuttal and discussion, main concerns included limited evaluation scope,  incomplete positioning w.r.t. prior work (e.g., hypernetworks, LoRA-based methods), and method complexity and design choices (e.g., mapper dependence, lack of symmetry handling). These issues were clarified to a sufficient extent and do not undermine the core contribution.

Overall, the paper is technically sound, clearly written, and makes a meaningful contribution to weight-space learning. Therefore my recommendation is accept.

For the revision, the authors should include all discussed edits, in particular:

- include all the additional experiments and baselines discussed,
- better position the method against prior work (especially hypernetworks and related weight-space approaches),
- clarify the intended use case and moderate claims around zero-shot generation,
- briefly discuss key design choices and limitations (e.g., symmetry handling).